# Update on the Diagnosis of Behçet’s Disease

**DOI:** 10.3390/diagnostics13010041

**Published:** 2022-12-23

**Authors:** Fatma Alibaz-Oner, Haner Direskeneli

**Affiliations:** Division of Rheumatology, Department of Internal Medicine, School of Medicine, Marmara University, 34722 Istanbul, Turkey

**Keywords:** Behçet’s disease, diagnosis, venous wall thickness, ultrasonography

## Abstract

Behçet’s disease (BD) is a systemic inflammatory disease with unknown etiology. It is characterized by recurrent mucocutaneous lesions and major organ disease such as ocular, neurologic, vascular, and gastrointestinal manifestations. The diagnosis of BD is mainly based on clinical manifestations after ruling out other potential causes. There are no specific laboratory, histopathologic, or genetic findings for the diagnosis of BD. The International Study Group (ISG) criteria set is still the most widely used set for the diagnosis. The main limitation of this criteria set is the lack of major organ manifestations such as vascular, neurologic, and gastrointestinal involvement. The ICBD 2014 criteria are more sensitive, especially in early disease. However, patients with such as spondyloarthritis can easily meet this criteria set, causing overdiagnosis. Diagnosing BD can be a big challenge in daily practice, especially in patients presenting with only major organ involvement such as posterior uveitis, neurologic, vascular, and gastrointestinal findings with or without oral ulcers. These patients do not meet ISG criteria and can be diagnosed with “expert opinion” in countries with high BD prevalence. The pathergy test is the only diagnostic test used as diagnostic or classification criteria for BD. Our recent studies showed that common femoral vein (CFV) thickness measurement can be a valuable, practical, and cheap diagnostic tool for BD with sensitivity and specificities higher than 80% for the cut-off value of 0.5 mm. However, the diagnostic accuracy of CFV measurement should be investigated in other disease groups in the differential diagnosis of BD and in also different ethnic populations.

## 1. Introduction

Behçet’s disease (BD) is a systemic inflammatory disease with unknown etiology. BD was defined as a variable-vessel vasculitis in the revised Chapel Hill Consensus Conference due to the involvement of both arterial and venous vessels of all sizes [1]. BD is first described with “triple symptom complex” as oral ulcers, genital ulcers, and uveitis by Professor Hulusi Behçet in 1937 [2]. It is characterized by recurrent mucocutaneous lesions and major organ involvement, such as ocular, neurologic, vascular, and gastrointestinal manifestations. BD has a distinct geographical distribution. It is more prevalent in countries around the Mediterranean basin and East Asia such as Japan, Korea and China (ancient silk road). The prevalence was reported to be 370/10,000 in Turkey, 11.9/10,000 in Israel, 13.5/10,000 in Japan, 0.64/10,000 in England, 5.2/10,000 in the United States, and 3.8–15.9/10,000 in Italy [3]. BD frequently starts in second and third decades of the life and has a remitting–relapsing course with decreasing disease activity during older age. Complete remission was observed in at least 60% of patients during 20 years of follow-up. The disease course is more severe in male BD patients, especially those younger than 25 years old [4].

The diagnosis of BD is mainly based on clinical manifestations after ruling out other potential causes. There are no specific laboratory, histopathologic, or genetic findings for the diagnosis of BD. Furthermore, there is a large geographical variation both in the disease prevalence and the disease manifestations. Therefore, the diagnosis of BD may be difficult in patients presenting with only major organ involvement such as posterior uveitis, neurologic, vascular, and gastrointestinal manifestations. The emergence of other disease manifestations aiding the definite diagnosis of BD can take months and even years in this group of patients. The disease can also remain limited in some patients, which causes diagnostic difficulty. In a recent study of 189 suspected or probable BD patients followed up for 3.2 years, only 71 (37.6%) patients were classified as BD according to the International Study Group (ISG) criteria during follow-up [5]. In fact, recent data highlight an increase in the frequency of incomplete BD in Far Eastern countries, such as Japan and Korea [6]. In this group of patients, diagnosis is made according to the presence of specific clinical manifestations of BD by ‘expert opinion’. The specific clinical findings such as genital ulcers, ocular, vascular, and parenchymal neurological involvement were proposed to be defined as strong elements by Yazici et al. for the differential diagnosis of Behçet’s disease [3]. The aim of this brief review is to discuss the new developments in the diagnosis of BD, mainly diagnostic criteria and tools, in light of the current data.

## 2. Clinical Manifestations of Behçet’s Disease

### 2.1. Mucocutaneous Manifestations

Recurrent aphthous (oral) ulcers are seen in 95–97% of patients and are usually the first disease manifestation of BD. It precedes the diagnosis by an average of 6–7 years [7] (Table 1). In long-term, routine follow-up, we observed that oral ulcers were the main cause of ongoing clinical activity in BD [8]. Genital ulcers are another major manifestation of BD. They are also the most specific (95%) mucocutaneous sign of BD and seen in 50–85% of BD patients. Genital ulcers are generally located on the scrotum in males and on the major and minor labiae in females. They leave scars in about half of the patients [9]. Papulopustular lesions are seen at usual acne sites, such as the face, upper chest, and back, and additionally on the legs and arms. They are generally indistinguishable from ordinary acne [10]. Erythema nodosum-like lesions are red, painful, erythematous non-ulcerating nodules that are frequently located on the legs. They are seen in approximately 50% of patients and more frequently present in females. Superficial thrombophlebitis is a frequent type of venous involvement. It presents as palpable, painful subcutaneous nodules that are string-like hardenings with reddening of the overlying skin [11]. In some cases, the differentiation of superficial thrombophlebitis and erythema nodosum-like lesions can be very difficult in daily practice.

### 2.2. Musculoskeletal Involvement

Arthritis or arthralgia is seen in about 50% of patients with BD. Musculoskeletal involvement of BD is a non-deforming, non-erosive peripheral oligoarthritis. Most frequently involved joints are the knees, ankles, hands, and wrists. It usually resolves in days or weeks. Sacroiliitis may be a part of the musculoskeletal involvement of BD in some cases [12].

### 2.3. Major Organ Involvement

Ocular involvement is one of the main causes of morbidity in BD. It is observed in up to 50% of patients, and is generally bilateral. Ocular inflammation is commonly panuveitis and retinitis. However, isolated anterior uveitis can be seen in some cases. At the end of a 20-year follow-up, ocular involvement was bilateral in 87% of males and 71% of females [13].

Vascular involvement is the one of the main causes of mortality and morbidity, especially in young males [4]. Vessels of all sizes can be involved, in both the arterial and venous systems, in the vascular involvement of BD [14]. Major vessel involvement types are vein thromboses, arterial occlusions, and arterial aneurysms. Venous involvement is reported to be more common than arterial involvement (up to 80%). Vascular involvement is observed in up to 40% of the patients with BD. Deep venous thrombosis in lower extremity is the most frequent form of vascular involvement [15]. Although venous thrombosis is seen primarily in lower extremities, it may also affect different sites, including the inferior and superior vena cava, pulmonary artery, suprahepatic vessels, and cardiac cavities. Up to 17% of the mortality in BD is reported to be associated with venous involvement such as pulmonary embolism or Budd–Chiari syndrome [16]. 

Neurologic involvement is seen in 5% of patients with BD. There are two main forms: vascular and parenchymal. Parenchymal neuro-BD leads to inflammatory lesions in the brain stem, diencephalon, basal ganglia and, less frequently, the spinal cord and cerebellum. It usually presents with bilateral pyramidal signs, unilateral hemiparesis, behavioral changes, sphincter disturbances, and headache. The vascular neuro-BD is dural sinus thrombosis, mainly characterized by headache and papilledema. It usually associates with venous thrombosis in lower extremities, and has better prognosis compared to parenchymal neuro-BD [17,18].

Gastrointestinal involvement is seen in one-third of BD patients in Japan and Korea, but it is rare in Mediterranean countries (<5%). Mucosal aphthous ulcers, primarily in the terminal ileum and the cecum, are characteristic signs of intestinal involvement of BD. The most frequent symptoms are vomiting, abdominal pain, and diarrhea [19]. Gastrointestinal involvement of BD and Crohn’s disease can be indistinguishable in some cases.

### 2.4. Other Clinical Findings

Fever is not a typical sign in BD. It can be seen in major vascular, neurological involvement, and arthritis [20]. Testicular pain and epididymitis can be observed in male patients [21]. In a study from Turkey, it was reported that incidence of varicocele was increased in Behcet’s disease [22]. Glomerulonephritis was rarely reported in BD. AA-type amyloidosis can occasionally be seen in longstanding disease [23].

## 3. Laboratory

There is no characteristic or pathognomonic laboratory finding in BD. Erythrocyte sedimentation rate and C-reactive protein levels are usually mildly elevated, mainly in cases with arthritis, erythema nodosum-like lesions, or vascular disease. Autoantibodies such as rheumatoid factor, antinuclear, anticardiolipin, and antineutrophil cytoplasmic antibodies are generally absent. However, it was reported that BD patients with gastrointestinal involvement had higher levels of anti-*Saccharomyces cerevisiae* antibodies compared to BD patients without gastrointestinal involvement [24]. BD has also no specific histopathologic features. The strongest genetic association between HLA-B51 and BD was first reported by Ohno et al. [25]. Then, it was shown in different ethnic populations to have positivity ranges around 40–60% in BD [26]. However, HLA-B51 has low diagnostic value for daily practice usage due to its high frequency in the general population in the countries with high BD prevalence [3]. The genome-wide association studies showed associations with interleukin (IL)-10 and IL23R-IL12RB2 loci, ERAP-1, CCR1-CCR3, KLRC4 and STAT4 [27,28]. 

## 4. Diagnostic and Classification Criteria Sets for Behçet’s Disease

The ISG criteria, which are the most widely used for diagnosis, were published in 1990 (Table 2) [29]. This criteria set was generated by a group of experts following large number of BD patients in daily practice. The presence of oral ulcers is accepted as sine qua non. Additionally, two of the following—genital ulceration, eye lesions, skin lesions, and positive pathergy test—are needed for diagnosis of BD. These criteria had been shown to have 95% sensitivity and 98% specificity. The main limitation of this criteria set is the exclusion of major organ involvement such as vascular, neurologic, and gastrointestinal involvement. Low positivity of the pathergy (skin prick) test, possibly related to less traumatic punctures and changing microbial skin flora, is another limitation of the ISG criteria set [30]. Fourteen years later, revised Japanese diagnostic criteria for BD were published in 2004. Clinical manifestations were divided into two groups of main and additional symptoms. Patients with four main symptoms were defined as complete BD; patients with three main symptoms, two main symptoms and two additional symptoms, typical ocular lesions with one main symptom or typical ocular lesions with two additional symptoms were defined as incomplete BD (Table 3) [31]. Although not widely accepted, the definition of incomplete BD led to the early diagnosis of patients presenting with limited disease manifestations, and possibly prevented the delay in the treatment. In 2014, international criteria for BD (ICBD) were published and included vascular and neurological involvement (Table 4) [32]. The ICBD criteria set is based on a scoring system attributing 2 points for oral ulcer, genital ulcer, and ocular lesions; 1 point for positive pathergy test, neurologic, and vascular involvement. Patients having ≥4 points are classified as BD. In a collaborative study of 27 countries, the ICBD criteria set demonstrated an unbiased estimate of sensitivity of 94.8%, which is considerably higher than that of the ISG criteria (85.0%). However, the specificity (90.5%) was lower compared to ISG (96%) [32]. However, the ICBD criteria, which seem more sensitive, especially in early disease, may cause overdiagnosis and patients with spondyloarthropathic features can be mislabeled as BD [3].

There are separate efforts for the diagnosis of specific disease manifestations of BD. In 2014, expert consensus recommendations were published for the diagnosis and management of Neuro-Behçet’s Disease. In the light of the clinical, laboratory, and neuroimaging findings, disease was defined as “definite” or “probable” Neuro-Behçet’s Disease. The novelty of these criteria is the definition of “probable disease” by expert consensus. According to these criteria, probable Neuro-Behçet’s disease was defined as the presence of typical neurological findings suggesting Neuro-Behçet’s disease without fulfillment of ISG criteria or non-characteristic neurological findings for Neuro-Behçet’s disease with the fulfillment of ISG criteria [33]. Although providing opportunity for the diagnosis of patients lacking typical features, this criteria set has not yet been validated. Recently, Tugal-Tutgun et al. reported an algorithm for the diagnosis of ocular involvement of Behçet disease in 2020. Superficial retinal infiltrate, signs of occlusive retinal vasculitis, and diffuse retinal capillary leakage, as well as the absence of granulomatous anterior uveitis or choroiditis in patients with vitritis were the items with the highest accuracy in classification and regression tree analysis [34]. These organ-specific diagnostic approaches can be helpful, especially for the diagnosis of patients presenting with only major organ involvement in daily practice.

For pediatric BD patients, classification criteria were proposed in 2016 by an international expert consensus group (PEDBD). This criteria set contains six categories (oral, genital, skin, ocular, neurological, and vascular involvement). At least three findings (each from a different category) are needed to define pediatric BD [35]. This new international PEDBD criteria set has higher sensitivity (91.7%) but lower specificity (42.9%) when compared to ISG in the pediatric population [36].

## 5. Pathergy Test

The skin pathergy reaction (SPR) is the only diagnostic test currently existing for BD. It is a nonspecific hyperinflammatory response to sterile needle-induced tissue damage, first described by Blobner [37] and Jensen [38]. Despite the lack of consensus on the methodology of pathergy testing, in total, 4–6 sharp or blunted needle pricks are performed. Twenty-gauge needles are inserted either perpendicularly or oblique through the glabrous skin of both forearms after cleaning with an antiseptic. Fresko et al. showed that surgical cleaning of the skin with disinfectants reduces the positivity rate of pathergy test in BD [39]. When 20 G and 26 G needles were tested in BD, higher positivity rate with 20G needles was acquired [40]. The positivity of SPR is defined as the presence of an erythematous papule ≥2 mm or a pustule after 24–48 h. The positivity is affected by many factors such as the usage of sharp or blunted needles, prick number, gender, and disease activity. There is also great geographic variation. The positivity rate is reported to range between 7.7 and 84%. While BD patients had a low positivity rate in Northern Europe, patients from Turkey, China, and Middle Eastern countries had higher positivity rates. SPR can also be positive in other diseases such as Sweet’s syndrome, Crohn’s disease, pyoderma gangrenosum, A20 haploinsufficiency, and a few others. Although SPR is quite specific for BD, the sensitivity decreased in the last few decades, possibly due to less traumatic punctures and changing microbial skin flora [41]. Davatchi et al. assessed the affect of the positivity of pathergy test on the 16 available classification/diagnosis criteria sets for Behcet’s disease. They found that the sensitivity and the accuracy decreased, the specificity increased without positive pathergy test in 15 out of 16 criteria sets [42]. Very recently, Deniz et al. reported that 23 valent polysaccharide pneumococcal vaccine application to the skin increased pathergy positivity both in active (sensitivity: 64.3%, specificity: 100%) and inactive BD patients (sensitivity: 80.3%, specificity: 100%) [43].

## 6. Femoral Vein Wall Thickness Measurement as a Diagnostic Tool for Behcet’s Disease 

Vascular involvement is seen in about one-third of BD patients [15]. Venous vessels are involved in 67–84% of all vascular manifestations [15,44]. Arterial vessels are involved in less than 15% of vascular BD patients [45]. Deep vein thrombosis (DVT) in the lower extremity is the most frequent form of vascular BD (around 80%). Despite the dominant venous involvement, the data investigating veins directly are very limited in BD. Few data pointing out the vein wall inflammation also in skin, ocular, and neurologic involvement of BD have been previously published [46,47,48,49].

The first study directly assessing veins in BD was reported by Ambrose N et al. In this magnetic resonance imaging study, increased vein wall thickness (VWT) was reported in popliteal veins of BD patients [50]. Boulon et al. later published a case presenting with acute calf pain (without thrombosis). Increased VWT in the right great saphenous vein was detected in this case by ultrasound (US) [51]. We recently published the first controlled doppler US study showing increased VWT of lower extremity veins in male BD patients independent of vascular involvement [52,53]. Among all thickened lower extremity veins, the common femoral vein (CFV), which is the largest vein of the lower extremity, was chosen as the primary site in the US assessment. Bilateral CFV measurements had high area under the receiver operating characteristic curves (>0.8) with sensitivities of 81–82.8% and specificities of 78.4–81.1% for the cut-off values of 0.5 mm. Positive (PPV) and negative (NPV) predictive values for this cut-off were also acceptable (PPV: 85.7–87.5%, NPV: 72.5–75%) in our study [52,53] (Figure 1). Other studies from Turkey reported increased VWT in BD, and confirmed our observations [54,55,56]. Recently, we assessed the wall thickness of the pulmonary artery, which is most frequently involved artery in BD. Pulmonary artery had a similar structure to systemic veins in terms of width and thin-walled vessels with increased compliance. We found that PA wall thickness is increased in patients with only major organ involvement, which can be a sign of more severe disease in BD patients [57].

We later assessed the *diagnostic performance* of CFV thickness measurement in BD compared with multiple disease control groups such as ankylosing spondylitis, systemic vasculitides, antiphospholipid syndrome, venous insufficiency, and non-inflammatory DVT. Our findings indicated an increased VWT, a distinctive feature of BD, rarely present in other inflammatory or vascular diseases. The cut-off value of ≥0.5 mm, determined in our first study, performed quite well against all control groups with sensitivity and specificity (with the exception of antiphospholipid syndrome) higher than >80%. We also found that values especially higher than 0.75 mm seem to indicate a very high probability of BD [58]. In our recent study, CFV measurement was also found to be a distinctive diagnostic tool for the differentiation of BD and Crohn’s disease, which is an important challenge in daily practice, in especially BD patients with GI involvement [59]. Most recently, increased venous wall thickness was found in childhood BD with and without vascular involvement, and Atalay et al. suggested that increased VWT may be a new criterion for the diagnosis in both definite and incomplete pediatric BD patients [60]. Taken together, assessing VWT seems to be a cheap, easy, and widely available tool aimed at the diagnosis of BD. 

## 7. Conclusions

The diagnosis of BD is mainly based on clinical manifestations after excluding other potential mimickers. There is no specific laboratory or genetic test. Diagnosing BD can be a big challenge in daily practice, especially in patients presenting with only major organ involvement with or without oral ulcers. The early diagnosis of BD has a critical value, especially in patients presenting with only vascular thrombosis to prevent the delay in the treatment. The treatment of vascular thrombosis in BD is different from noninflammatory DVT, as it needs immunosuppressive treatment rather than anticoagulants. Until recently, the pathergy test was the only diagnostic test that was used as diagnostic or classification criteria for BD. Our recent studies showed that CFV thickness measurement can be a valuable, practical, and cheap diagnostic tool for BD with the sensitivity and the specificities higher than 80% for the cut-off value of 0.5 mm. However, the diagnostic accuracy of CFV measurement should be investigated in other disease groups in the differential diagnosis of BD and in different ethnic populations.

## Figures and Tables

**Figure 1 diagnostics-13-00041-f001:**
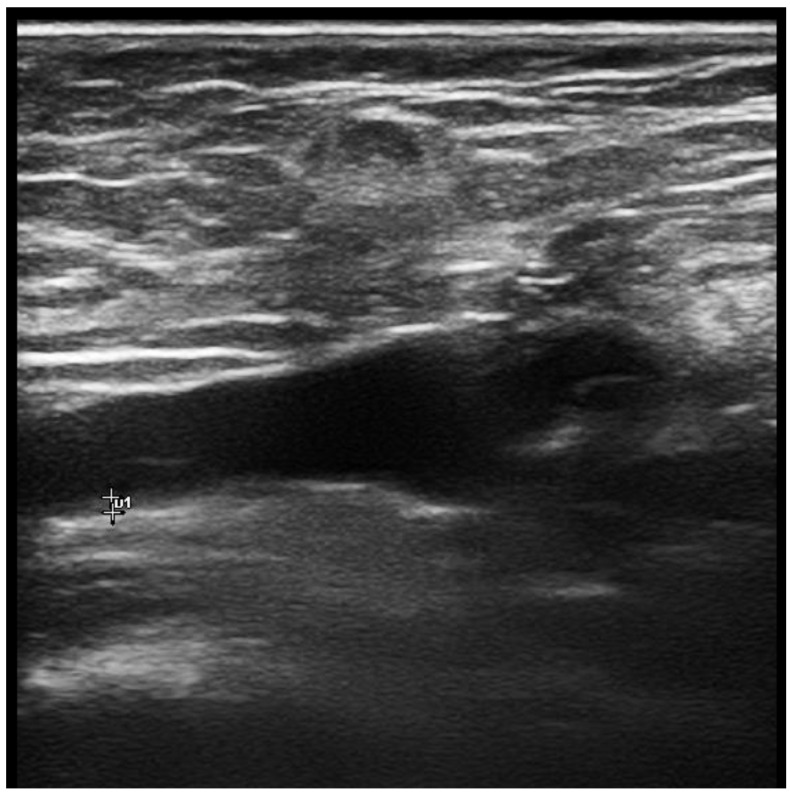
Measurement of common femoral vein thickness by ultrasonography.

**Table 1 diagnostics-13-00041-t001:** Frequency of Behçet’s disease manifestations.

Manifestation	Frequency (%)
Oral ulcers	97–99
Genital ulcers	85
Genital scar	50
Papulopustular lesions	85
Erythema nodosum	50
Pathergy reaction	40–60
Uveitis	50
Arthritis	30–50
Subcutaneous thrombophlebitis	25
Deep vein thrombosis	15
Arterial occlusion/aneurysm	5–10
Central nervous system involvement	20
Epididymitis	5
Gastrointestinal lesions	1–58

**Table 2 diagnostics-13-00041-t002:** International Study Group criteria for the diagnosis of Behçet’s disease.

Manifestation	Definition
Recurrent oral ulceration	Observed by a physician or reported reliably by patient, recurring at least three times in one 12-month period
**Plus any two of the following findings:**	
Recurrent genital ulceration	Recurrent genital aphthous ulceration or scarring, observed by a physician or reported reliably by patient
Eye lesions	Anterior uveitis, posterior uveitis, or cells in vitreous on slit lamp examination; or retinal vasculitis observed by qualified physician (ophthalmologist)
Skin lesions	Erythema nodosum-like lesions, observed by a physician or reported reliably by patient; Pseudofolliculitis or papulopustular lesions; or acneiform nodules observed by a physician in post adolescent patients not receiving glucocorticoids
Positive pathergy test	Test interpreted as positive by a physician at 24–48 h, performed with oblique insertion of a 20-gauge needle or smaller under sterile conditions

**Table 3 diagnostics-13-00041-t003:** Revised diagnostic criteria proposed by the Behcet’s Disease (BD) Research Committee of Japan.

**Main Symptoms**
Recurrent aphthous ulcers on oral mucosa Skin lesions a. Skin lesion with erythema nodosum b. Subcutaneous thrombophlebitis c. Follicular papules, acneiform papules c. cf.) Skin hypersensitivity Ocular lesions a. Iridocyclitis b. Posterior uveitis (retinochoroiditis) c. If the patients have the following eye symptoms after (a) and (b), diagnose as BD lesions in accordance with (a) and (b) c. Posterior adhesion of iris, pigmentation on lens, retinochoroid atrophy, atrophy of optic nerve, complicated cataract, secondary c. glaucoma, leakage of bulbus oculi Genital ulcer
**Additional symptoms**
Arthritis without deformity or sclerosis Epididymitis Gastrointestinal lesion represented by ileocecal ulceration Vascular lesions Central nervous system lesions, moderate or severe
**Criteria for diagnosis of disease types**
*Complete type:* The four main symptoms appeared during the clinical course *Incomplete types:* Three of the main four symptoms, or two main symptoms and two additional symptoms, appeared during the clinical course Typical ocular lesion and another main symptom, or two additional symptoms appeared during the clinical course *BD suspected:* Although some main symptoms appear, the case does not meet the criteria for the incomplete type Typical additional symptom is recurrent or becomes more severe *Special lesions:* Gastrointestinal lesions—presence of abdominal pain and occult blood should be confirmed Vascular lesions—vasculitis of aorta, artery, large veins, or small veins should be differentially diagnosed Neuronal lesions—presence of headache, paresis, lesions of brain and spinal cord, mental symptoms, and other symptoms should be confirmed

**Table 4 diagnostics-13-00041-t004:** International criteria for Behçet’s disease.

BD Manifestations	Score Assigned
Ocular lesions	2
Oral aphthosis	2
Genital aphthosis	2
Skin lesions	1
Neurological manifestations	1
Vascular manifestations	1
Positive pathergy test *	1

Patients are classified as having BD with scores ≥4. * Pathergy test is optional.

## Data Availability

Not applicable.

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
