# Peer review of "Update on the Diagnosis of Behçet’s Disease"

_diagnostics, 2022, doi:10.3390/diagnostics13010041_

Round 1
Reviewer 1 Report
General Comment:
The review is well structured and highlights an important update about CFV thickness measurement in patients with Behcet's disease. Authors have conducted many studies prior on CFV thickness measurement in these patients with comparing healthy and disease controls. Although I want to say that this diagnostic tool requires validation globally before including it as a diagnostic tool in patients with Behcet's disease.
There should be information about HLA B51 and B5 as a utility tool in diagnosing BD. Authors have not mentioned it.
Few lines about Genes and histopathology too should be added in the laboratory section.
Introduction: Very well written, but Chappel Hill consensus conference was held in 2012, so its not recent and this word can be removed.
Clinical Manifestations of Behcet's: Well written and can also add a summarising table with frequency of each manifestations.
Laboratory: Correct reactive spelling (ractive). The summary of all criteria well written, but its better to give both sensitivity and specificity of each criteria and their limitations. Can make a comparative table of all these criteria in place of individual table which will help readers to make a note. Also can add the terminology O' Duffy Goldstein criteria in Neuro Behcet section.
Pathergy test can be performed with 20-25G needles, so it can be added.
CFV thickness: As already mentioned, its a newer technique that shows promise but has to be replicated in other studies as well. Also requires validation in patients at risk or incomplete Behcet's or patients with non vascular involvement. Are there ant=y similar studies from across the globe from other authors which have shown similar findings? If yes, plz mention.
Thank You
Author Response
Reviewer 1:
General Comment:
The review is well structured and highlights an important update about CFV thickness measurement in patients with Behcet's disease. Authors have conducted many studies prior on CFV thickness measurement in these patients with comparing healthy and disease controls. Although I want to say that this diagnostic tool requires validation globally before including it as a diagnostic tool in patients with Behcet's disease.
Response: I agree on that a diagnostic tool in patients with Behcet's disease needs globally validation. Our multi-ethnic validation study is going on.
There should be information about HLA B51 and B5 as a utility tool in diagnosing BD. Authors have not mentioned it. Few lines about Genes and histopathology too should be added in the laboratory section.
Response: Added into manuscript and highlighted.
Introduction: Very well written, but Chappel Hill consensus conference was held in 2012, so its not recent and this word can be removed.
Response: Recent Word is removed as suggested.
Clinical Manifestations of Behcet's: Well written and can also add a summarising table with frequency of each manifestations.
Response: A table for clinical manifetstaions were added into the manuscript anf highlighted.
Laboratory: Correct reactive spelling (ractive). The summary of all criteria well written, but its better to give both sensitivity and specificity of each criteria and their limitations. Can make a comparative table of all these criteria in place of individual table which will help readers to make a note. Also can add the terminology O' Duffy Goldstein criteria in Neuro Behcet section.
Response: Manuscript was revised according to the suggestions and highlighted.
Pathergy test can be performed with 20-25G needles, so it can be added.
Response: Different sized needles comparison data was added inthe manuscript and highlighted.
CFV thickness: As already mentioned, its a newer technique that shows promise but has to be replicated in other studies as well. Also requires validation in patients at risk or incomplete Behcet's or patients with non vascular involvement. Are there ant=y similar studies from across the globe from other authors which have shown similar findings? If yes, plz mention.
Response: Our results were confirmed in 3 other studies conducted in adults from Turkey, also confirmed in pediatric onset both in definite and incomplete BD. These studies were cited in the manuscript, and highlighted. Also, our multi-ethnic validation study is now going-on.
Reviewer 2 Report
Thank you for a comprehensive review on BD.
Just one point needs clarification, the procedure of Pathergy test, should the site be cleaned with antiseptic? Is it going to lead to a false positive pathergy test?
Author Response
Reviewer 2:
Thank you for a comprehensive review on BD.
Just one point needs clarification, the procedure of Pathergy test, should the site be cleaned with antiseptic? Is it going to lead to a false positive pathergy test?
Response: The pathergy test should be performed on skin after cleaning with an antiseptic. This was clarified in the manuscript. Also, it was shown that use of disinfectants reduce the positivity of pathergy test, ıt was added in the text

Round 2
Reviewer 1 Report
Dear Author,
The revised version of the manuscript is well written and structured and potentially hints at another investigational technique which might help in detecting cases of Behcets disease, although it needs validation by others as well.
All the points mentioned previously are well covered.
Conclusion seems to be too long and the initial paragraph seems to be repetition of introduction, needs to have a clear and short message.
Author Response
Reviewer 1:
The revised version of the manuscript is well written and structured and potentially hints at another investigational technique which might help in detecting cases of Behcets disease, although it needs validation by others as well.
All the points mentioned previously are well covered.
Conclusion seems to be too long and the initial paragraph seems to be repetition of introduction, needs to have a clear and short message.
Response: Conclusion was shortened as suggested reviewer 1.
